# CEIP: Combining Explicit and Implicit Priors for Reinforcement Learning with Demonstrations

Kai Yan        Alexander G. Schwing        Yu-Xiong Wang
University of Illinois Urbana-Champaign
{kaiyan3, aschwing, yxw}@illinois.edu
https://github.com/289371298/CEIP

## Abstract

Although reinforcement learning has found widespread use in dense reward settings, training autonomous agents with sparse rewards remains challenging. To address this difficulty, prior work has shown promising results when using not only task-specific demonstrations but also task-agnostic albeit somewhat related demonstrations. In most cases, the available demonstrations are distilled into an implicit prior, commonly represented via a single deep net. Explicit priors in the form of a database that can be queried have also been shown to lead to encouraging results. To better benefit from available demonstrations, we develop a method to Combine Explicit and Implicit Priors (CEIP). CEIP exploits multiple implicit priors in the form of normalizing flows in parallel to form a single complex prior. Moreover, CEIP uses an effective explicit retrieval and push-forward mechanism to condition the implicit priors. In three challenging environments, we find the proposed CEIP method to improve upon sophisticated state-of-the-art techniques.

## 1 Introduction

Reinforcement learning (RL) has found widespread use across domains from robotics [57] and game AI [44] to recommender systems [6]. Despite its success, reinforcement learning is also known to be sample inefficient. For instance, training a robot arm with sparse rewards to sort objects from scratch still requires many training steps if it is at all feasible [46].

To increase the sample efficiency of reinforcement learning, prior work aims to leverage demonstrations [4, 34, 40]. These demonstrations can be *task-specific* [4, 17], i.e., they directly correspond to and address the task of interest. More recently, the use of *task-agnostic* demonstrations has also been studied [14, 16, 34, 46], showing that demonstrations for loosely related tasks can enhance sample efficiency of reinforcement learning agents.

To benefit from either of these two types of demonstrations, most work distills the information within the demonstrations into an *implicit prior*, by encoding available demonstrations in a deep net. For example, SKiLD [34] and FIST [16] use a variational auto-encoder (VAE) to encode the "skills," i.e., action sequences, in a latent space, and train a prior conditioned on states based on demonstrations to use the skills. Differently, PARROT [46] adopts a state-conditional normalizing flow to encode a transformation from a latent space to the actual action space. However, the idea of using the available demonstrations as an *explicit prior* has not received a lot of attention. Explicit priors enable the agent to maintain a database of demonstrations, which can be used to retrieve state-action sequences given an agent's current state. This technique has been utilized in robotics [32, 47] and early attempts of reinforcement learning with demonstrations [4]. It was also implemented as a baseline in [14]. One notable recent exception is FIST [16], which queries a database of demonstrations using the current state to retrieve a likely next state. The use of an explicit prior was shown to greatly enhance the

performance. However, FIST uses pure imitation learning without any RL, hence losing the chance for trial and remedy if the imitation is not good enough.

Our key insight is to leverage demonstrations both explicitly *and* implicitly, thus benefiting from both worlds. To achieve this, we develop **CEIP**, a method which **c**ombines **e**xplicit and **i**mplicit **p**riors. **CEIP** leverages implicit demonstrations by learning a transformation from a latent space to the real action space via normalizing flows. More importantly, different from prior work, such as PARROT and FIST which combine all the information within a single deep net, **CEIP** selects the most useful prior by combining multiple flows *in parallel* to form a single large flow. To benefit from demonstrations explicitly, **CEIP** augments the input of the normalizing flow with a likely future state, which is retrieved via a lookup from a database of transitions. For an effective retrieval, we propose a push-forward technique which ensures the database to return future states that have not been referred to yet, encouraging the agent to complete the whole trajectory even if it fails on a single task.

We evaluate the proposed approach on three challenging environments: fetchreach [36], kitchen [11], and office [45]. In each environment, we study the use of both task-specific and task-agnostic demonstrations. We observe that integrating an explicit prior, especially with our proposed push-forward technique, greatly improves results. Notably, the proposed approach works well on sophisticated long-horizon robotics tasks with a few, or sometimes even one task-specific demonstration.

## 2 Preliminaries

**Reinforcement Learning.** Reinforcement learning (RL) aims to train an *agent* to make the 'best' decision towards completing a particular task in a given environment. The environment and the task are often described as a Markov Decision Process (MDP), which is defined by a tuple $(\mathcal{S}, \mathcal{A}, T, r, \gamma)$. In timestep $t$ of the Markov process, the agent observes the current *state* $s_t \in \mathcal{S}$, and executes an *action* $a_t \in \mathcal{A}$ following some probability distribution, i.e., *policy* $\pi(a_t|s_t) \in \Delta(\mathcal{A})$, where $\Delta(\mathcal{A})$ denotes the probability simplex over elements in space $\mathcal{A}$. Upon executing action $a_t$, the state of the agent changes to $s_{t+1}$ following the dynamics of the environment, which are governed by the *transition function* $T(s_t, a_t) : \mathcal{S} \times \mathcal{A} \to \Delta(\mathcal{S})$. Meanwhile, the agent receives a *reward* $r(s_t, a_t) \in \mathbb{R}$. The agent aims to maximize the cumulative reward $\sum_t \gamma^t r(s_t, a_t)$, where $\gamma \in [0, 1]$ is the discount factor. One complete run in an environment is called an *episode*, and the corresponding state-action pairs $\tau = \{(s_1, a_1), (s_2, a_2), \dots\}$ form a *trajectory $\tau$*.

**Normalizing Flows.** A normalizing flow [24] is a generative model that transforms elements $z_0$ drawn from a simple distribution $p_z$, e.g., a Gaussian, to elements $a_0$ drawn from a more complex distribution $p_a$. For this transformation, a bijective function $f$ is used, i.e., $a_0 = f(z_0)$. The use of a bijective function ensures that the log-likelihood of the more complex distribution at any point is tractable and that samples of such a distribution can be easily generated by taking samples from the simple distribution and pushing them through the flow. Formally, the core idea of a normalizing flow can be summarized via $p_a(a_0) = p_z(f^{-1}(a_0)) \log \left| \frac{\partial f^{-1}(a)}{\partial a}|_{a=a_0} \right|$, where $|\cdot|$ is the determinant (guaranteed positive by flow designs), $a$ is a random variable with the desired more complex distribution, and $z$ is a random variable governed by a simple distribution. To efficiently compute the determinant of the Jacobian matrix of $f^{-1}$, special constraints are imposed on the form of $f$. For example, coupling flows like RealNVP [8] and autoregressive flows [31] impose the Jacobian of $f^{-1}$ to be triangular.

## 3 CEIP: Combining Explicit and Implicit Priors

### 3.1 Overview

As illustrated in Fig. 1, our goal is to train an autonomous agent to solve challenging tasks despite sparse rewards, such as controlling a robot arm to complete item manipulation tasks (like turning on a switch or opening a cabinet). For this we aim to benefit from available demonstrations. Formally, we consider a task-specific dataset $D_{\text{TS}} = \{\tau_1^{\text{TS}}, \tau_2^{\text{TS}}, \dots, \tau_m^{\text{TS}}\}$, where $\tau_i^{\text{TS}}$ is the $i$-th trajectory of the task-specific dataset, and a task-agnostic dataset $D_{\text{TA}} = \{\bigcup D_i | i \in \{1, 2, 3, \dots, n\}\}$, where $D_i = \{\tau_1^i, \tau_2^i, \dots, \tau_{m_i}^i\}$ subsumes the demonstration trajectories for the $i$-th task in the task-agnostic dataset. Each trajectory $\tau = \{(s_1, a_1), (s_2, a_2), \dots\}$ in the dataset is a state-action pair sequence of a complete episode, where $s$ is the state, and $a$ is the action. We assume that the number of

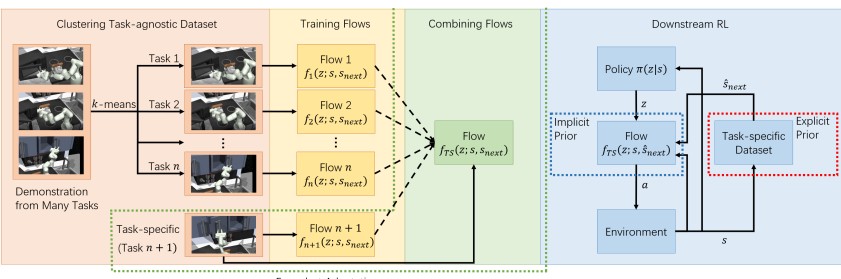

Figure 1: Overview of our proposed approach, CEIP. Our approach can be divided into three steps: a) cluster the task-agnostic dataset into different tasks, and then train one flow on each of the $n$ tasks of the task-agnostic dataset; b) train a flow on the task-specific dataset, and then train the coefficients to combine the $n + 1$ flows into one large flow $f_{\text{TS}}$, which is the implicit prior; c) conduct reinforcement learning on the target task; for each timestep, we perform a dataset lookup in the task-specific dataset to find the state most similar to current state $s$, and return the likely *next state* $\hat{s}_{\text{next}}$ in the trajectory, which is the explicit prior.

available task-specific trajectories is very small, i.e., $\sum_{i=1}^{n} m_i \gg m$, which is common in practice. For readability, we will also refer to $D_{\text{TS}}$ using $D_{n+1}$.

Our approach leverages demonstrations implicitly by training a normalizing flow $f_{\text{TS}}$, which transforms the probability distribution represented by a policy $\pi(z|s)$ over a simple latent probability space $\mathcal{Z}$, i.e., $z \in \mathcal{Z}$, into a reasonable expert policy over the space of real-world actions $\mathcal{A}$. As before, $s$ is the current environment state. Thus, the downstream RL agent only needs to learn a policy $\pi(z|s)$ that results in a probability distribution over latent space $\mathcal{Z}$, which is subsequently mapped via the flow $f_{\text{TS}}$ to a real-world action $a \in \mathcal{A}$. Intuitively, the MDP in the latent space is governed by a less complex probability distribution, making it easier to train because the flow increases the exposure of more likely actions, while reducing the chance that a less-likely action is chosen. This is because the flow reduces the probability mass for less likely actions given the current state.

Task-agnostic demonstrations contain useful patterns that may be related to the task at hand. However, not all the task-agnostic data are always equally useful, as different task-agnostic data may require to expose different parts of the action space. Therefore, different from prior work where all data are fed into the same deep net model, we first partition the task-agnostic dataset into different groups according to task similarity so as to increase flexibility. For this we use a classical $k$-means algorithm. We then train different flows $f_i$ on each of the groups, and finally combine the flows via learned coefficients into a single flow $f_{\text{TS}}$. Beneficially, this process permits to expose different parts of the action space as needed and according to perceived task similarity.

Lastly, our approach further leverages demonstrations explicitly, by conditioning the flow not only on the current state but also on a likely next state, to better inform the agent of the state it should try to achieve with its current action. In the following, we first discuss the implicit prior of **CEIP** in Sec. 3.2; afterward we discuss our explicit prior in Sec. 3.3, and the downstream reinforcement learning with both priors in Sec. 3.4.

## 3.2 Implicit Prior

To better benefit from demonstrations implicitly, we use a 1-layer normalizing flow as the backbone of our implicit prior. It essentially corresponds to a conditioned affine transformation of a Gaussian distribution. We choose a flow-based model instead of a VAE-based one for two reasons: 1) as the dimensionality before and after the transformation via a normalizing flow remains identical and since the flow is invertible, the agent is guaranteed to have control over the whole action space. This ensures that all parts of the action space are accessible, which is not guaranteed by VAE-based methods like SKiLD or FIST; 2) normalizing flows, especially coupling flows such as RealNVP [8], can be easily stacked *horizontally*, so that the combination of parallel flows is also a flow. Among feasible flow models, we found that the simplest 1-layer flow suffices to achieve good results, and is even more robust in training than a more complex RealNVP. Next, in Sec. 3.2.1 we first introduce details regarding the normalizing flow $f_i$, before we discuss in Sec. 3.2.2 how to combine the flows into one flow $f_{\text{TS}}$ applicable to the task for which the task-specific dataset contains demonstrations.

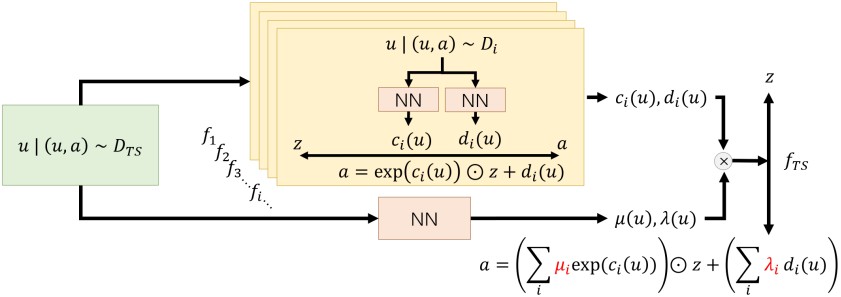

Figure 2: An illustration of how we combine different flows into one large flow for the task-specific dataset. Each red block of "NN" stands for a neural network. Note that $c_i(u)$ and $d_i(u)$ are vectors, while $\mu_i$ and $\lambda_i$ are the $i$-th dimension of $\mu(u)$ and $\lambda(u)$.

**3.2.1 Normalizing Flow Prior.** For each task $i$ in the task-agnostic dataset, i.e., for each $D_i$, we train a conditional 1-layer normalizing flow $f_i(z; u) = a$ which maps a latent space variable $z \in \mathbb{R}^q$ to an action $a \in \mathbb{R}^q$, where $q$ is the number of dimensions of the real-valued action vector. We let $u$ refer to a conditioning variable. In our case $u$ is either the current environment state $s$ (if no explicit prior is used) or a concatenation of the current and a likely next state $[s, s_{\text{next}}]$ (if an explicit prior is used). Concretely, the formulation of our 1-layer flow is

$$f_i(z; u) = a = \exp\{c_i(u)\} \odot z + d_i(u), \tag{1}$$

where $c_i(u) \in \mathbb{R}^q$, $d_i(u) \in \mathbb{R}^q$ are trainable deep nets, and $\odot$ refers to the Hadamard product. The $\exp$ function is applied elementwise. When training the flow, we sample state-action pairs (without explicit prior) or transitions (with explicit prior) $(u, a)$ from the dataset $D_i$, and maximize the log-likelihood $\mathbb{E}_{(u,a) \sim D_i} \log p(a|u)$; refer to [24] for how to maximize this objective.

In the discussion above, we assume the decomposition of the task-agnostic dataset into tasks to be given. If such a decomposition is not provided (e.g., for the kitchen and office environments in our experiments), we perform a $k$-means clustering to divide the task-agnostic dataset into different parts. The clustering algorithm operates on the last state of a trajectory, which is used to represent the whole trajectory. The intuition is two-fold. First, for many real-world MDPs, achieving a particular terminal state is more important than the actions taken [12]. For example, when we control a robot to pick and place items, we want all target items to reach the right place eventually; however, we do not care too much about the actions taken to achieve this state. Second, among all the states, the final state is often the most informative about the task that the agent has completed. The number of clusters $k$ in the $k$-means algorithm is a hyperparameter, which empirically should be larger than the number of dimensions of the action space. Though we assume the task-agnostic dataset is partitioned into labeled clusters, our experiments show that our approach is robust and good results are achieved even without a precise ground-truth decomposition.

In addition to the clusters in the task-agnostic dataset, we train a flow $f_{n+1}(z; u) = a$ on the task-specific dataset $D_{n+1} = D_{\text{TS}}$, using the same maximum log-likelihood loss, which is optional but always available. This is not necessary when the task is relatively simple and the episodes are short (e.g., the fetchreach environment in the experiment section), but becomes particularly helpful in scenarios where some subtasks of a task sequence only appear in the task-specific dataset (e.g., the kitchen environment).

**3.2.2 Few-shot Adaptation.** The flow models discussed in Sec. 3.2.1 learn which parts of the action space to be more strongly exposed from the latent space. However, not all the flows expose useful parts of the action space for the current state. For example, the target task needs the agent to move its gripper upwards at a particular location, but in the task-agnostic dataset, the robot more often moves the gripper downwards to finish another task. In order to select the most useful prior, we need to tune our set of flows learned on the task-agnostic datasets to the small number of trajectories available in the task-specific dataset. To ensure that this does not lead to overfitting as only a very small number of task-specific trajectories are available, we train a set of coefficients that selects the flow that works the best for the current task. Concretely, given all the trained flows, we train a set of coefficients to combine the flows $f_1$ to $f_n$ trained on the task-agnostic data, and also the flow $f_{n+1}$ trained on the task-specific data. The coefficients select from the set of available flows the most useful one. To

achieve this, we use the combination flow illustrated in Fig. 2 which is formally specified as follows:

$$f_{\text{TS}}(z; u) = \left( \sum_{i=1}^{n+1} \mu_i(u) \exp\{c_i(u)\} \right) \odot z + \left( \sum_{i=1}^{n+1} \lambda_i(u) d_i(u) \right). \tag{2}$$

Here, $\mu_i(u) \in \mathbb{R}$, $\lambda_i(u) \in \mathbb{R}$ are the $i$-th entry of the deep nets $\mu(u) \in \mathbb{R}^{n+1}$, $\lambda(u) \in \mathbb{R}^{n+1}$, respectively, which yield the coefficients while the deep nets $c_i$ and $d_i$ are frozen. As before, the exp function is applied elementwise. We use a softplus activation and an offset at the output of $\mu$ to force $\mu_i(u) \geq 10^{-4}$ for any $i$ for numerical stability. Note that the combined flow $f_{\text{TS}}$ consisting of multiple 1-layer flows is also a 1-layer normalizing flow. Hence, all the compelling properties over VAE-based architectures described at the beginning of Sec. 3.2 remain valid. To train the combined flow, we use the same log likelihood loss $\mathbb{E}_{(u,a)\sim D_{\text{TS}}} \log p(a|u)$ as that for training single flows. Here, we optimize the deep nets $\mu(u)$ and $\lambda(u)$ which parameterize $f_{\text{TS}}$.

Obviously, the employed combination of flows can be straightforwardly extended to a more complicated flow, e.g., a RealNVP [8] or Glow [22]. However, we found the discussed simple formulation to work remarkably well and to be robust.

## 3.3 Explicit Prior

Beyond distilling information from demonstrations into deep nets which are then used as implicit priors, we find explicit use of demonstrations to also be remarkably useful. To benefit, we encode future state information into the input of the flow. More specifically, instead of sampling $(s, a)$-pairs from a dataset $D$ for training the flows, we consider sampling a *transition* $(s, a, s_{\text{next}})$ from $D$. During training, we concatenate $s$ and $s_{\text{next}}$ before feeding it into a flow, i.e., $u = [s, s_{\text{next}}]$ instead of $u = s$.

However, we do not know the future state $s_{\text{next}}$ when deploying the policy. To obtain an estimate, we use task-specific demonstrations as explicit priors. More formally, we use the trajectories within the task-specific dataset $D_{\text{TS}}$ as a database. This is manageable as we assume the task-specific dataset to be small. For each environment step of reinforcement learning with current state $s$, we perform a lookup, where $s$ is the query, states $s_{\text{key}}$ in the trajectories are the keys, and their corresponding next state $s_{\text{next}}$ is the value. Concretely, we assume $s_{\text{next}}$ belongs to trajectory $\tau$ in the task-specific dataset $D_{\text{TS}}$, and define $\hat{s}_{\text{next}}$ as the result of the database retrieval with respect to the given query $s$, i.e.,

$$\hat{s}_{\text{next}} = \text{argmin}_{s_{\text{next}}|(s_{\text{key}},a,s_{\text{next}})\in D_{\text{TS}}} [(s_{\text{key}} - s)^2 + C \cdot \delta(s_{\text{next}})], \text{ where}$$

$$\delta(s_{\text{next}}) = \begin{cases} 1 \text{ if } \exists s'_{\text{next}} \in \tau, \text{ s.t. } s'_{\text{next}} \text{ is no earlier than } s_{\text{next}} \text{ in } \tau \text{ and has been retrieved,} \\ 0 \text{ otherwise.} \end{cases} \tag{3}$$

In Eq. (3), $C$ is a constant and $\delta$ is the indicator function. We set $u = [s, \hat{s}_{\text{next}}]$ as the condition, feed it into the trained flow $f_{\text{TS}}$, and map the latent space element $z$ obtained from the RL policy to the real-world action $a$. The penalty term $\delta$ is a push-forward technique, which aims to push the agent to move forward instead of staying put, imposing monotonicity on the retrieved $\hat{s}_{\text{next}}$. Consider an agent at a particular state $s$ and a flow $f_{\text{TS}}$, conditioned on $u = [s, \hat{s}_{\text{next}}]$ which maps the chosen action $z$ to a real-world action $a$ that does not modify the environment. Without the penalty term, the agent will remain at the same state, retrieve the same likely next state, which again maps onto the action that does not change the environment. Intuitively, this term discourages 1) retrieving the same state twice, and 2) returning to earlier states in a given trajectory. In our experiments, we set $C = 1$.

## 3.4 Reinforcement Learning with Priors

Given the implicit and explicit priors, we use RL to train a policy $\pi(z|s)$ to accomplish the target task demonstrated in the task-specific dataset. As shown in Fig. 1, the RL agent receives a state $s$ and provides a latent space element $z$. The conditioning variable of the flow is retrieved via the dataset lookup described in Sec. 3.3 and the real-world action $a$ is then computed using the flow. Note, our approach is suitable for any RL method, i.e., the policy $\pi(z|s)$ can be trained using any RL algorithm such as proximal policy optimization (PPO) [43] or soft-actor-critic (SAC) [15].

## 4 Experiments

In this section, we evaluate our CEIP approach on three challenging environments: fetchreach (Sec. 4.1), kitchen (Sec. 4.2), and office (Sec. 4.3), which are all tasks that manipulate a robot arm.

In each experiment, we study the following questions: 1) Can the algorithm make good use of the demonstrations compared to baselines? 2) Are our core design decisions (e.g., state augmentation with explicit prior and the push-forward technique) indeed helpful?

**Baselines.** We compare the proposed method to three baselines: PARROT [46], SKiLD [34], and FIST [16]. In all environments, we use reward as our criteria (higher is better), and the results are averaged over 3 runs for SKiLD (which is much slower to train than other baselines) and 9 runs for all other methods unless otherwise mentioned. To differentiate variants of the PARROT baseline and our method, we use suffixes. We use "EX" to refer to variants that use an explicit prior, and "forward" to denote variants with the push-forward technique. For our method, if we train a task-specific flow on $D_{\text{TS}} = D_{n+1}$, we append the abbreviation "TS." For PARROT, the use of the task-specific data is indicated with "TS" and the use of task-agnostic data is indicated with "TA."[1] See Table 3 and Table 4 for precise correspondence.

## 4.1 FetchReach Environment

**Environment Setup.** The agent needs to control a robot arm to move its gripper to a goal location in 3D space, and remain there. During an episode of $40$ steps, the agent receives a 10-dimensional state about its location and outputs a 4-dimensional action, which indicates the change of coordinates of the agent and the openness of the gripper. It will receive a reward of $0$ if it arrives and stays in the vicinity of its target. Otherwise, it will receive a reward of $-1$. This environment is a harder version of the FetchReach-v1 robotics environment in gym [36], where we increase the average distance of the starting point to the goal, effectively increasing the training difficulty. Moreover, to test the robustness of the algorithm, we sample a random action from a normal distribution at the beginning of each episode, which the agent executes for $x$ steps before the episode begins. We use $x \sim U[5, 20]$. For simplicity, we denote the goal generated with azimuth $\frac{\pi d}{4}$ as "direction $d$" (e.g., direction 4.5).

**Dataset Setup.** We use trajectories from directions $d \in \{0, 1, \ldots, 7\}$ as the task-agnostic data. Each task includes $40$ trajectories, and each of the trajectories has $40$ steps, i.e., $1600$ environment steps in total. The task-specific datasets contain directions $4.5, 5.5, 6.5$, and $7.5$. (The robot cannot reach the other four .5 directions due to physical limits.) For each task-specific dataset, we use $4$ trajectories, for a total of $160$ environment steps.

**Experimental Setup.** For fetchreach, we use a fully-connected deep net with one hidden layer of width $32$ and ReLU [1] activation as a standard "block" of our algorithm (each block corresponds to a red "NN" rectangle in Fig. 2). We have a pair of blocks for $c_i(s)$ and $d_i(s)$ for each flow $f_i$. For flow training, we train $8$ flows for $8$ directions in the task-agnostic dataset without the explicit prior. We use a batch size of $40$ and train for $1000$ epochs for both each flow and the combination of flows, with gradient clipping at norm $10^{-4}$, learning rate $0.001$, and Adam optimizer [21]. We use the model that has the best performance on the validation dataset at the end of every epoch. For each dataset, we randomly draw $80\%$ state-action pairs (or transitions in ablation) as the training set and $20\%$ state-action pairs as the validation set. The combination of flow is also a block, which outputs both $\mu(s)$ and $\lambda(s)$. See the Appendix for the implementation details of SKiLD, FIST, and PARROT. For each method with RL training, we use a soft-actor-critic (SAC) [15] with 30K environment steps, a batch size of $256$, and $1000$ steps of initial random exploration. Unless otherwise noted, all other RL hyperparameters in all experiments use the default values of Stable-baselines3 [38].

**Main Results.** Fig. 3 shows the results for different methods without explicit priors or task-specific single flow $f_{n+1}$. In all four tasks, our method significantly outperforms the other baselines. This indicates that the flow training indeed helps boost the exploration process. Naïve reinforcement learning from scratch fails in most cases, which underscores the necessity of utilizing demonstrations to aid RL exploration. As this is a simple task with only a few wildly varied trajectories, adding a flow for the task-specific dataset does not improve our method. Noteworthy, neither SKiLD nor FIST works on fetchreach. Their VAE-based architecture with each action sequence as the agent's output ("skill") can not be trained with the little amount of wildly varied data with short horizon. Flow-based models like ours and PARROT, which only consider the action of the current step instead of the action sequence, work better.

---

[1] The original PARROT in [46] is essentially PARROT+TA. It is straightforward to use PARROT directly on the task-specific dataset. Hence, we tried PARROT+TS and PARROT+(TS+TA) as well.

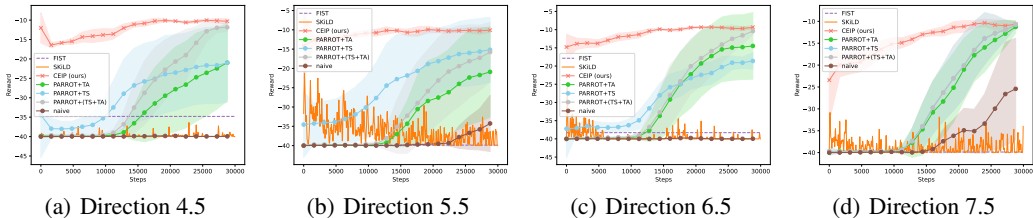

|   |   |   |   |
|---|---|---|---|
| (a) Direction 4.5 | (b) Direction 5.5 | (c) Direction 6.5 | (d) Direction 7.5 |

Figure 3: Main performance results on the fetchreach environment for different directions, where the lines are the mean reward (higher is better) and shades are the standard deviation. FIST is represented by a dashed line as it does not require RL.

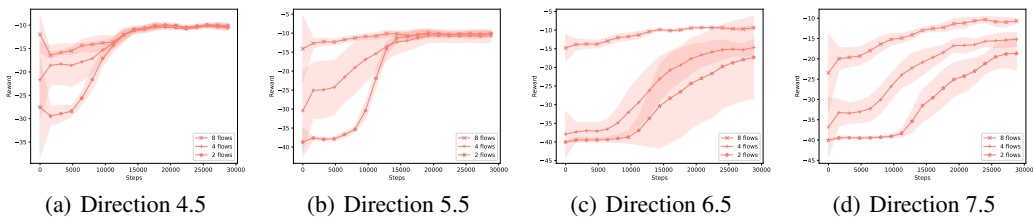

|   |   |   |   |
|---|---|---|---|
| (a) Direction 4.5 | (b) Direction 5.5 | (c) Direction 6.5 | (d) Direction 7.5 |

Figure 4: Ablation on the number of flows used in our method. We observe more flows to lead to better performance, likely because expressivity increases which helps in fitting the expert policy.

**Are more flows helpful for CEIP?** Fig. 4 shows the performance of our method using a different number of flows, which are trained on the data of the directions that are the closest to the task-specific direction (e.g., directions 5 and 6 for 2 flows with the target being direction 5.5). The result shows that within a reasonable range, increasing the number of flows improves the expressivity and consequently results of our model. See Appendix D for more ablation studies.

## 4.2 Kitchen Environment

**Environment Setup.** We use the kitchen environment adopted from D4RL [11], which serves as a testbed for many reinforcement learning and imitation learning approaches, like SPiRL [33], SKiLD [34], relay policy learning [14], and FIST [16]. The agent needs to control a 7-DOF robot arm to complete a sequence of four tasks (e.g., move the kettle, or slide the cabinet) in the correct order. The agent will receive a $+1$ reward only upon finishing a task, and 0 otherwise. The action space is 9-dimensional and the state space is 60-dimensional. This environment is very challenging, as high-precision control of the robot arm is needed and also a long horizon of 280 timesteps is needed. Moreover, there is a small noise applied to each agent action, which requires the agent to be robust.

**Dataset Setup.** We use two dataset settings, which are adopted from SKiLD and FIST (denoted as *Kitchen-SKiLD* and *Kitchen-FIST* below). In Kitchen-SKiLD, we use 601 teleoperated sequences that perform a variety of task sequences as the task-agnostic dataset, and use *only one* trajectory for the task-specific dataset. In Kitchen-FIST, we use part of the task-agnostic dataset (about $200 - 300$ trajectories) that *does not* contain a particular task in the task-specific dataset, and use *only one* trajectory for the task-specific dataset. There are two different task-specific datasets in Kitchen-SKiLD, and four different task-specific datasets in Kitchen-FIST. The latter is significantly harder, as the agent must learn a new task from very little data. For simplicity, we denote them as "SKiLD-A/B" and "FIST-A/B/C/D" respectively. See Appendix C for details on each task.

**Experimental Setup.** We use $k$-means to partition the task-agnostic datasets into 24 different clusters, and train 24 flows accordingly. For each flow, we use a fully-connected network with 2 hidden layers of width 256 with ReLU activation as a "block" for our algorithm. For the combination of flows, we use a fully-connected network with 1 hidden layer of width 64 with ReLU activation. Each layer of the deep net (except the output layer) described above has a 1D batchnorm function. The blocks are used analogously to the fetchreach environment. We use a batch size of 256 for the task-agnostic dataset and a batch size of 128 for the task-specific dataset. Other training hyperparameters are identical to the fetchreach environment. For each RL training, we use proximal policy optimization

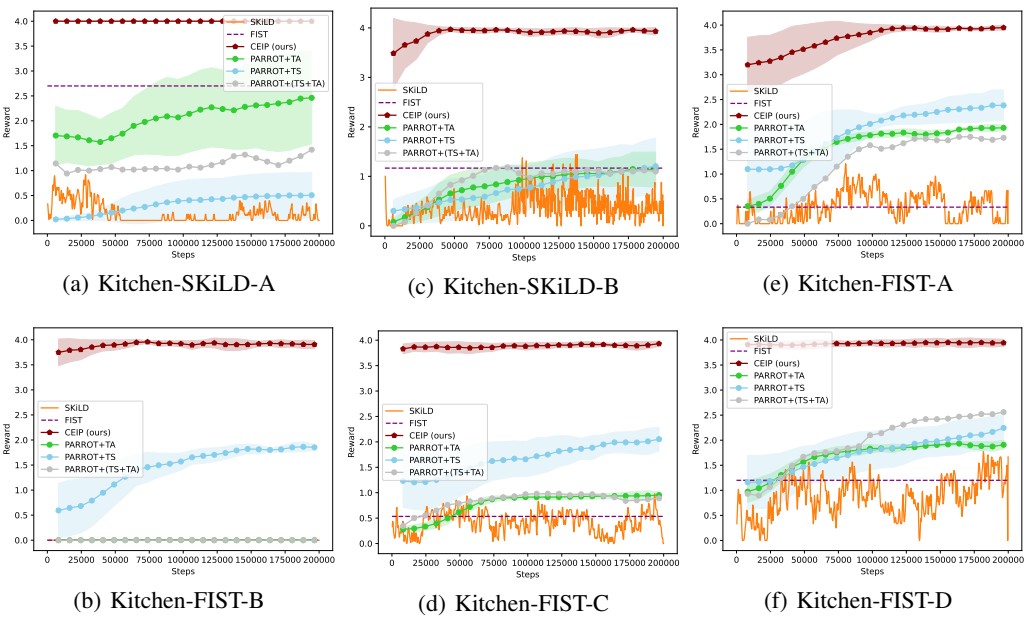

Figure 5: Comparison on Kitchen-SKiLD and Kitchen-FIST environments.

(PPO) [43] for 200K environment steps, with update interval being $2048$ (Kitchen-SKiLD) / $4096$ (Kitchen-FIST), 60 epochs per update, and a batch size of $64$.

**Main Results.** Fig. 5 shows the main results on Kitchen-SKiLD and Kitchen-FIST. Our method outperforms all other baselines in all of the 6 settings of the task-agnostic and task-specific datasets. For our method, we use the task-specific single flow $f_{n+1}$, explicit prior, and the push-forward technique. We compare to the original PARROT formulation. See Appendix D for ablation studies of PARROT with explicit prior and our method without $f_{n+1}$ or explicit prior.

**Does CEIP overly rely on the task-specific flow if it is used?** One concern for our method could be: does the task-specific single flow dominate the model? Theoretically, when all flows are perfect, a trivial combination of flows that minimizes the training objective is to set $\lambda_{n+1} = 1, \mu_{n+1} = 1$ for the task-specific single flow, and $\lambda_i \approx 0, \mu_i \approx 0$ for $i \neq n+1$. To study this concern, we plot the change of the coefficient $\mu$ during an episode in Fig. 6. We observe that the single flow on the task-specific dataset is not dominating the combination of the flow, despite being trained on the task-specific dataset. The blue curve with legend 'TA-8' in Fig. 6 shows the coefficient for the 8th flow trained in the task-agnostic dataset. It exhibits an increase of $\mu$ at the end of an episode, as the last subtask in the target task is more relevant to the prior encoded in the 8th flow. Intuitively, over-reliance in our design (Fig. 8 in the Appendix) is discouraged, because of the softplus function and the positive offset applied on $\mu$. For over-reliance, all task-agnostic flows $f_i$ with $i \in \{1, 2, \ldots, n\}$ should have a coefficient of $\mu_i = 0$, which is hard to approach due to the offset of $\mu$ and softplus. In fact, a degenerated CEIP is essentially PARROT+TS(+EX+forward), which is worse than our method but still a powerful baseline.

**Is reinforcement learning useful in cases with a perfect initial reward?** Fig. 6 shows the episode length of our method on Kitchen-SKiLD-A. Even if the reward is already perfect, the reinforcement learning process is still able to maximize discounted reward, which optimizes the path.

### 4.3 Office Environment

**Environment and Dataset Setups.** We follow SKiLD, where a robot with 8-dimensional action space and 97-dimensional state space needs to put three randomly selected items on a table into three containers in the correct, randomly generated order. The agent will receive a $+1$ reward when it completes a subtask (e.g., picking up an item, or dropping the item at the right place), and $0$ otherwise. This environment is even harder than the kitchen environment, as the agent must manipulate freely movable objects and the number of possible subtasks in the task-agnostic dataset is much larger than that in the kitchen environment. We use the same task-agnostic dataset as SKiLD, which contains

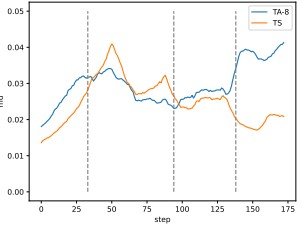

(a) Illustration of coefficient

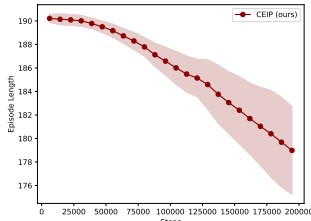

(b) Average episode length

Figure 6: a) Illustration of the coefficient change of a trained CEIP model during an episode of Kitchen-SKiLD-A. This CEIP model is trained with the task-specific single flow and without the explicit prior. 'TA-8' is the 8-th single flow for the task-agnostic dataset, and 'TS' is the single flow for the task-specific dataset. The grey dotted lines are the partition of different subtasks. b) Average episode length of our method on the Kitchen-SKiLD-A task. The episode ends immediately when all the tasks are completed; thus, shortening length means that RL helps to find policy with more efficient completion of tasks.

2400 trajectories with randomized subtasks sampled from a script policy. For the task-specific dataset, we use 5 trajectories for a particular combination of tasks.

**Experimental Setup.** Similar to the kitchen environment, we use $k$-means over the last state of each trajectory and partition the task-agnostic dataset into 24 clusters. The architecture and training paradigm of the flow model are identical to those used in the kitchen environment. For RL training, we use PPO for 2M environment steps, with update interval being 4096 environment steps, 60 epochs per update, and a batch size of 64. All other hyperparameters follow the kitchen environment setting. We run each method with 3 different seeds.

**Main Results and Ablation.** Fig. 7a shows the main result across different methods. Our method with explicit prior, push-forward technique, and task-specific flow outperforms all baselines. FIST works well in this environment, probably because of two reasons: 1) there are a sufficient number of

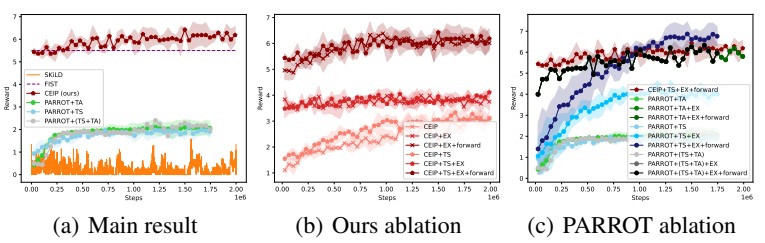

(a) Main result     (b) Ours ablation     (c) PARROT ablation

Figure 7: Main result and ablation of our method and PARROT on the office environment.

task-specific trajectories for the VAE-architecture, and 2) the office environment is less noisy than the kitchen environment. However, as FIST does not contain a reinforcement learning stage, it has no chance to improve on a decent policy which could have been a good start for an RL agent. Fig. 7b shows the ablation of our method. While the task-specific single flow $f_{n+1}$ does not help in this environment, the explicit prior greatly improves results. Also, as illustrated, the reward curve of the variants with the explicit prior but without the push-forward skill does not grow, which is due to the agent getting stuck as described at the end of Sec. 3.3. Fig. 7c shows the ablation result of PARROT, which also emphasizes that the explicit prior and push-forward skill greatly improve results.

# 5 Related Work

**Reinforcement Learning with Demonstrations.** Using demonstrations to improve the sample efficiency of RL is an established direction [13, 26, 41, 55]. Recently, the use of task-agnostic demonstrations has gained popularity, as task-specific data need to be sampled from a particular expert and can be expensive to acquire [33, 34, 46]. To utilize the prior, skill-based methods such as SPiRL [33], SKiLD [34], and SIMPL [30] extract action sequences from the dataset with a VAE-based model, while TRAIL [58] recovers transitions from a task-agnostic dataset with uniformly randomly sampled action. Our method considers the situation where both task-agnostic *and* task-specific data exist and significantly improves results over prior work with similar settings, e.g., SKiLD [34].

**Action Priors.** An action prior is a common way to utilize demonstrations for reinforcement learning [34] and imitation learning [16]. Most work uses an implicit prior, where a probability distribution of actions conditioned on a state is learned by a deep net and then used to rule out unlikely attempts [3], to form a hierarchical structure [34, 46], or to serve as a regularizer for RL training [34, 40], preventing the agent to stray too far from expert demonstrations. Explicit priors are less explored. They come in the form of nearest neighbors [2] (as in our work) or in the form of locally weighted regression [32]. They are utilized in robotics [2, 32, 42, 47] and early work of RL with demonstrations [4]. Another way to explicitly use demonstrations includes filling the buffer of offline RL algorithms with transitions sampled from an expert dataset to help exploration [17, 29, 54]. Different from all such work, we propose a novel way of using both implicit and explicit priors.

**Normalizing Flow.** Normalizing flows are a generative model that can be used for variational inference [23, 52] and density estimation [18, 31] and come in different forms: RealNVP [8], Glow [22], or autoregressive flow [31]. Many methods use normalizing flows in reinforcement learning [20, 27, 28, 48, 49, 56] and imitation learning [5]. However, most prior work uses normalizing flows as a strong density estimator to exploit a richer class of policies. Most closely related to our work is PARROT [46], which trains a single normalizing flow as an implicit prior. Different from our work, PARROT does not differentiate tasks among the task-agnostic dataset and does not use an explicit prior. More importantly, different from prior work, we develop a simple yet effective way to combine flows using learned coefficients. While there are some approaches that combine flows via variational mixtures [7, 35], they have not been shown to succeed on challenging RL tasks.

**Few-shot Generalization.** Few-shot generalization [50] is broadly related, as a model is first trained across different datasets, and then adapted to a new dataset with small sample size. For example, similar to our work, FLUTE [51], SUR [9], and URT [25] use models for multiple datasets, which are then combined via weights for few-shot adaptation. Other methods have shared parameters across different tasks and only used some components within the model for adaptation [10, 37, 39, 51, 59]. While most work focuses on classification tasks, we address more complex RL tasks. Also, different from existing work, we found training of independent 1-layer flows without shared layers to be more flexible, and free from negative transfer as also reported by [19].

## 6 Discussion and Conclusion

We developed **CEIP**, a method for reinforcement learning which combines explicit and implicit priors obtained from task-agnostic and task-specific demonstrations. For implicit priors we use normalizing flows. For explicit priors we use a database lookup with a push-forward retrieval. In three challenging environments, we show that **CEIP** improves upon baselines.

**Limitations.** Limitations of CEIP are as follows: 1) *Training time*. The use of demonstrations requires training a decent number of flows which can be time-consuming, albeit mitigated to some extent by parallel training. 2) *Reliance on optimality of expert demonstrations.* Similar to prior work like SKiLD [34] and FIST [16], our method assumes availability of optimal state-action trajectories for the target task. Accuracy of those demonstrations impacts results. Future work will focus on improving robustness and generality. 3) *Balance between the degree of freedom and generalization in fitting the flow mixture.* Fig. 4 reveals that more degrees of freedom in the flow mixture improve results of CEIP. Our current design uses a linear combination which offers $O(n)$ degrees of freedom ($\mu$ and $\lambda$), where $n$ is the number of flows. In contrast, too many degrees of freedom will result in overfitting. It is interesting future work to study this tradeoff.

**Societal impact.** Our work helps to train RL agents more efficiently from demonstrations for the same and closely related tasks, particularly when the environment only provides sparse rewards. If successful, this expands the applicability of automation. However, increased automation may also cause job loss which negatively impacts society.

**Acknowledgements.** This work was supported in part by NSF under Grants 1718221, 2008387, 2045586, 2106825, MRI 1725729, NIFA award 2020-67021-32799, the Jump ARCHES endowment through the Health Care Engineering Systems Center, the National Center for Supercomputing Applications (NCSA) at the University of Illinois at Urbana-Champaign through the NCSA Fellows program, and the IBM-Illinois Discovery Accelerator Institute. We thank NVIDIA for a GPU.

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
