# OpenReview forum: "CEIP: Combining Explicit and Implicit Priors for Reinforcement Learning with Demonstrations"
_NeurIPS.cc/2022/Conference — NeurIPS 2022 Accept_

### Official Review · Reviewer_P2zU · 2022-06-25

**Rating:** 7
**Confidence:** 4
**Soundness:** 3 good
**Presentation:** 4 excellent
**Contribution:** 2 fair

**Summary:**

The paper proposes a method CEIP that leverages explicit and implicit priors from the demonstration for reinforcement learning. On top of PARROT [45] which uses normalizing flow prior to task agnostic dataset to generate action, CEIP learns coefficients to combine flows to generate one task-specific flow parallel and use (state, predicted next state) pairs as the input of the flow instead of the state. As a result, CEIP achieves better performance than its baselines, PARROT, FIST, and SKiLD.

**Questions:**

1. In Line 202, us -> use

2. In Figure 5 (b) and Figure 9, it is hard to differentiate between solid and dotted lines. It is better to use different markers for the same category of EX instead of two lines.

3. In Lines 164-167, “most useful” is a vague term, it would be better to use other expressions such as making the best performance, etc.


**Limitations:**

The authors addressed the limitations and potential negative societal impact of their work. As mentioned in the paper, I am concerned about the training time. I recommend parallelizing the training procedure.


**Strengths And Weaknesses:**

**[Strength]**

1. The paper is well-written and easy to understand.

2. Especially, the detailed description of baselines, environments and other experimental detail make the reader have a better understanding of with the attached codebase.

3. The proposed method clearly outperforms other baselines, PARROT, SKiLD, and FIST in all experiments and the authors also present the contribution of explicit prior by presenting Figure 5(c).


**[Weakness]**

1. It seems that there is no difference between task-agnostic flow (f_1 to f_n) and task-specific flow f_{i+1}. The ratio and size of the task-agnostic dataset and task-specific dataset are needed. Moreover, I recommend presenting the coefficient of f_{i+1} in those cases.

2. Another concern is that a detailed description difference between CEIP and PARROT-TS-EX-forward is missing. According to Line 220, PARROT-TS only uses task-specific data while CEIP-TS uses both task-specific and task-agnostic data. Is the dataset size a key factor of the difference or the Few-Shot Adaptation?


3. To present the effectiveness of the explicit prior, showing the results of TA-EX, TA-EA-forward in PARROT are better than TS-EX and TS-EX-forward (Figure 5(c)).


4. It is good to compare using CEIP with the ground-truth task label (oracle) for not using k-means clustering.

---

> ### Author Response · Authors · 2022-08-02
> **Response (1/2)**
>
> Thanks for appreciating our work.
>
> ### Q1: The ratio and size of the dataset; and the coefficient of f_{n+1}.
>
> As the reviewer correctly stated, there is no difference between the task-agnostic flow (f_1 to f_n) and the task-specific flow (f_{n+1}). We listed the ratio and size of the task-agnostic (TA) and task-specific (TS) data in the paper:
>
> For fetchreach (L695-696): TA data contains 1600 * 8 = 12800 state-action pairs; TS data contains 40 * 4 = 160 state-action pairs
>
> For kitchen-SKiLD (L709-712): TA data contains 136950 state-action pairs; TS data contains 214 (kitchen-SKiLD-A) or 262 (kitchen-SKiLD-B) state-action pairs
>
> For kitchen-FIST (Tab. 2 in the appendix): TA data contains around 45K to 65K state-action pairs; TS data contains around 200-250 state action pairs
>
> For office (L727-732): TA data contains 456033 state-action pairs; TS data contains 1155 state-action pairs
>
> Regarding f_{n+1}: we study the coefficient in Fig. 17 and L783-793.
>
> ### Q2a: Difference between CEIP and PARROT+TS+EX+forward
>
> Thanks for the suggestion. CEIP and PARROT+TS+EX+forward differ in CEIP’s use of a flow mixture trained on both task-specific and task-agnostic data. In contrast, PARROT+TS+EX+forward uses a single flow. We’ll clarify.
>
>
> ### Q2b: Is the dataset size, i.e. whether using both task-specific and task-agnostic data, key for the result difference?
>
> No, it is not key by itself. A combination of factors leads to improvements. In Fig. 11, we show that PARROT+(TS+TA) is not the best choice among all variants of PARROT. We also conduct experiments for PARROT+(TS+TA) with explicit prior and push-forward in Tab. C1 and C2 to better understand the effect of dataset size. The result shows that 1) dataset size alone makes no difference, and 2) the combination of dataset size and explicit prior improves PARROT, but with both components PARROT is still worse than CEIP and converges slower.
>
> For both tables, we list CEIP+TS+EX+forward (CEIP with task-specific flow, explicit prior and push-forward) for convenience.
>
> **Table C1:** Results of PARROT+(TS+TA) at beginning of RL
>
> Env | PARROT+(TS+TA) | PARROT+(TS+TA)+EX | PARROT+(TS+TA)+EX+forward | CEIP+TS+EX+forward
> ---|---|---|---|---
> Kitchen-SKiLD-A | 1.29 | 4 | 4 | 4
> Kitchen-SKiLD-B | 0 | 2.75 | 2.75 | 3.32
> Kitchen-FIST-A | 0 | 2.97 | 2.29 | 3.24
> Kitchen-FIST-B | 0 | 2.4 | 2.41 | 3.75
> Kitchen-FIST-C | 0.5 |  2.14 | 3.5 | 3.85
> Kitchen-FIST-D | 1 | 3.8 | 3.94 | 3.9
> Office | 0 | 4.32  | 4.32 | 3.55
>
> **Table C2:** Results of PARROT+(TS+TA) at end of RL
>
> Env | PARROT+(TS+TA) | PARROT+(TS+TA)+EX | PARROT+(TS+TA)+EX+forward | CEIP+TS+EX+forward
> ---|---|---|---|---
> Kitchen-SKiLD-A | 1.56 | 4 | 4 | 4
> Kitchen-SKiLD-B | 1.07 | 3.82  | 3.82 | 3.93
> Kitchen-FIST-A | 1.77 | 3 | 3.77 | 3.95
> Kitchen-FIST-B | 0 | 3.98 | 3.94 | 3.89
> Kitchen-FIST-C | 0.93 |  3.85 | 3.9 | 3.92
> Kitchen-FIST-D | 2.55 | 3.99 | 3.99 | 3.94
> Office | 0 | 5.89 | 5.89 | 6.33
>
> ### Q3: How well do TA+EX and TA+EX+forward do?
>
> Not as good as CEIP and PARROT+(TA+TS) with explicit prior and push-forward. The results are listed in Tab. C3 and C4. For both tables, we list CEIP+TS+EX+forward for convenience.
>
> **Table C3:** Results of PARROT+TA+EX and PARROT+TA+EX+forward at beginning of RL
>
> Env | PARROT+TA+EX | PARROT+TA+EX+forward | CEIP+TS+EX+forward
> ---|---|---|---
> Kitchen-SKiLD-A | 1.14 | 1.14 | 4
> Kitchen-SKiLD-B | 0.71 | 0.71 | 3.32
> Kitchen-FIST-A | 2.94 | 2.36 | 3.24
> Kitchen-FIST-B | 0 | 0  | 3.75
> Kitchen-FIST-C | 2.43 | 2 | 3.85
> Kitchen-FIST-D | 3 | 3 | 3.9
> Office | 4.32 | 4.32 | 3.55
>
> **Table C4:** Results of PARROT+TA+EX and PARROT+TA+EX+forward at end of RL
>
> Env | PARROT+TA+EX | PARROT+TA+EX+forward | CEIP+TS+EX+forward
> ---|---|---|---
> Kitchen-SKiLD-A | 3.86 | 3.86 | 4
> Kitchen-SKiLD-B | 2.43 | 2.43 | 3.93
> Kitchen-FIST-A | 3 | 2.98 | 3.95
> Kitchen-FIST-B | 0 | 0 | 3.89
> Kitchen-FIST-C | 2.85 | 2.68 | 3.92
> Kitchen-FIST-D | 3 | 3 | 3.94
> Office | 5.89 | 5.89 | 6.33
>
> The effectiveness of explicit prior and pushforward is shown in Tab. C3 and C4 as PARROT+TA+EX and PARROT+TA+EX+forward work much better than PARROT+TA. The effectiveness of using task-specific data is also shown by comparing Tab. C3, C4 to Tab. C1, C2: PARROT+TA+EX is generally worse than PARROT+(TA+TS)+EX.
>
> Note 1: PARROT+TA+EX and PARROT+TA+EX+forward have the same reward for some entries, which means PARROT+TA gets stuck less (as discussed in L196-200).
>
> Note 2: In Kitchen-FIST the third / first / second / third task is missing from the task-agnostic data for A / B / C / D, respectively, thus the reward for FIST-B is 0 and the others are limited correspondingly.

---

> > ### Comment · Reviewer_P2zU · 2022-08-03
> > **Concerns from the Author's Response**
> >
> > Thank the authors for dealing with the comments. I have several concerns and a comment regarding the response.
> >
> > **[Push Forward]**
> >
> > According to Figure 5b in the main paper, push forward heuristics which force the agent to move forward is a key factor to boost performance in CEIP. Moreover, as shown in Table 5c, it is also effective in PARROT. However, in Table 2, if PARROT also uses task-agnostic data, push-forward does not make any changes. Could you explain the reason?
> >
> > **[Office Result in Table 3]**
> >
> > Why CEIP+TA+EX+Push is significantly worse than others in Office in Table C3? The author should explain and analyze the reason that does not match with results in other environments.
> >
> >
> > **[GT24]**
> >
> > The authors mention that GT24 is to make the same number of ground truth that is used in CEIP by merging and splitting the ground truth depending on the environment. Could you describe the details of how to merge or split it?
> >
> >
> > **[Add Figure in the Revised Manuscript]**
> >
> > Presenting the tables in the response is appreciated. However, since this NeurIPS, paper change is allowed, it would be much better to add the Figure in the paper and refer it to here makes reviewers better understand.

---

> > > ### Author Response · Authors · 2022-08-06
> > > **Response for Further Concerns**
> > >
> > > ### Q1: Push Forward
> > >
> > > Push forward in PARROT+TA does not make a difference in Kitchen-SKiLD-A, Kitchen-SKiLD-B and office because, in these environments, PARROT+TA never reversed to earlier states in the expert trajectory. Thus push-forward does not take effect and does not affect the RL process. Note, this does not mean that PARROT+TA is closest to the expert trajectory, as the distance between a PARROT+TA state and the expert state can be, and is indeed, very large (for FIST [1] in their own experiment setting and CEIP+TS+EX+forward, the distance is usually < 0.01; for PARROT+TA, the distance often diverges and can be greater than 1). It only means that PARROT+TA moves in about the same direction as the expert trajectory (such that it will not refer to the same state twice), albeit with offsets.
> > >
> > > Push forward in Kitchen-FIST-B does not change results, because neither PARROT+TA+EX nor PARROT+TA+EX+forward learns to finish the first task – note, this first task is missing in the task-agnostic dataset. This prevents PARROT from picking up this skill.
> > >
> > > ### Q2: Office Result in rebuttal Table C3
> > >
> > > We apologize for the confusion. We made a mistake in rebuttal Tab. C3 and used CEIP+TS+EX’s initial reward for CEIP+TS+EX+forward, which makes CEIP+TS+EX+forward look worse than PARROT+TA+EX and PARROT+TA+EX+forward.  The correct initial reward for CEIP+TS+EX+forward should be 5.36. (See Fig. 5b for approximate initial reward for CEIP+TS+EX and CEIP+TS+EX+forward.) The results are consistent with those in other experiments.
> > >
> > > ### Q3: GT24
> > >
> > > The details are as follows:
> > >
> > > **Merging:** For Kitchen-SKiLD where the number of ground-truth labels is 33, there are exactly 9 labels that have no more than 3 demonstrations. We merge each of them into the label that is next to them in the dictionary order of concatenated task names.
> > >
> > > **Splitting:** For Kitchen-FIST where the number of ground-truth labels is x, x<24, we select the 24-x labels with the most demonstrations, and divide them evenly into two halves; each half is a new label. Note, no information is taken into account.
> > >
> > > We have included the details and corresponding experiments in Appendix D.3.
> > >
> > > ### Q4: Add figure in the revised Manuscript
> > >
> > > Thanks a lot for the advice. We have modified the figures in the paper and added the curves of the new experiments. All revisions by the reviewers’ advice are marked blue in the updated pdf. The figures that have been modified include Fig. 3, Fig. 4, Fig. 5, Fig. 9, Fig. 10, Fig. 11, Fig. 12 a) c) e), Fig. 13 b) d) g), Fig. 15 and Fig. 16, for which all captions are marked blue.
> > >
> > > **References**
> > >
> > > [1] K. Hakhamaneshi et al. Hierarchical few-shot imitation with skill transition models. ICLR, 2022.

---

> > > > ### Comment · Reviewer_P2zU · 2022-08-09
> > > > **Re: Response for Further Concerns, Keep Rating**
> > > >
> > > > Thank you for the clarification. I will keep my rating as a result of the author-reviewer discussion.

---

> ### Author Response · Authors · 2022-08-02
> **Response (2/2)**
>
> ### Q4: Comparing CEIP using k-means and ground-truth labeling
>
> We list the results in Tab. C5 and C6. Note we use 24 labels in k-means, but not all task-agnostic datasets have 24 ground truth labels. For a fair comparison, we also show the result using ground truth but pruned to 24 labels; we merge labels with few trajectories (for kitchen-SKiLD) and split labels with more trajectories (for kitchen-FIST) for this.
>
> Here are the meanings of each method in Tab. C5 and C6:
>
> **NFW:** No pushForWard (CEIP+TS+EX)
>
> **FW:** pushForWard (CEIP+TS+EX+forward)
>
> **GT24:** Ground Truth labels, but with merge and split to form 24 labels
>
> **GT:** Ground Truth labels; the number of subtasks differs
>
> **KM:** K-Means labels
>
> **Table C5:** Comparison between ground-truth label and k-means label for CEIP+TS+EX and CEIP+TS+EX+forward before RL.
>
> Environment | NFW+GT | FW+GT | NFW+GT24 | FW+GT24 | NFW+KM | FW+KM
> ---|---|---|---|---|---|---
> Kitchen-SKiLD-A | 4 | 4 | 4 | 4 | 4 | 4
> Kitchen-SKiLD-B | 3.96 | 3.95 | 4 | 4 | 3.81 | 3.32
> Kitchen-FIST-A | 3.59 | 3 | 3.68 | 3.24 | 3.44 | 3.41
> Kitchen-FIST-B | 4 | 4 | 3.76 | 3.75 | 3.8 | 4
> Kitchen-FIST-C | 3.94 | 3.81 | 3.80 | 3.85 | 4 | 3.94
> Kitchen-FIST-D | 3.6 | 3.4 | 3.96 | 3.9 | 3.75 | 3.76
>
> **Table C6:** Comparison between ground-truth label and k-means label for CEIP+TS+EX and CEIP+TS+EX+forward after RL.
>
> Environment | NFW+GT | FW+GT | NFW+GT24 | FW+GT24 | NFW+KM | FW+KM
> ---|---|---|---|---|---|---
> Kitchen-SKiLD-A | 4 | 4 | 4 | 4 | 4 | 4
> Kitchen-SKiLD-B | 3.87 | 3.87 | 4 | 4 | 4 | 4
> Kitchen-FIST-A | 3.93 | 3.9 | 3.92 | 3.99 | 3.94 | 3.95
> Kitchen-FIST-B | 3.8 | 3.74 | 3.97 | 3.88 | 3.92 | 3.89
> Kitchen-FIST-C | 3.94 | 3.96 | 3.99 | 3.95 | 3.93 | 3.92
> Kitchen-FIST-D | 3.71 | 3.93 | 3.87 | 3.96 | 3.95 | 3.94
>
> For kitchen-SKiLD, ground truth (both 24 flows and 33 flows) label works better than k-means label (Tab. C5 shows higher reward). For kitchen-FIST, the reward is similar before and after RL training, suggesting that the precise label doesn’t matter.
>
> The office environment contains 210 different tasks in the task-agnostic data, making it hard to train using the ground truth label.
>
> ### Q5: Typos, language and plotting
>
> Thanks for the suggestions, we’ll fix as suggested.
>
> ### Q6: Parallelization of training process
>
> We’ll parallelize training in our code release as suggested.

---

### Official Review · Reviewer_sUgF · 2022-07-11

**Rating:** 6
**Confidence:** 4
**Soundness:** 3 good
**Presentation:** 3 good
**Contribution:** 2 fair

**Summary:**

This submission proposes a skill learning algorithm for offline RL that can leverage task-specific demonstrations as implicit and explicit priors. The skill policy consists of a mixture of normalizing flows (trained on task-agnostic demonstrations) where the mixture weights are fine-tuned on task-specific demonstrations. The flows resemble an inverse model, in that they are conditioned on both the current and next state. The next state forms the explicit prior, as it is queried from the task-specific dataset via nearest-neighbor lookup for each state encountered during training. The explicit prior enables very effective few-shot generalization (i.e., utilizing the task-specific demonstrations without with random latent vectors, as this is what I suppose corresponds to the first data point in the learning curves), and learning a high-level policy to predict latent state flow inputs can further improve performance.

**Questions:**

- How would simply replaying a task-specific demonstration perform on the benchmarks? Does it already provide optimal reward and trajectories? I assume that fine-tuning the normalizing flow mixture requires ground truth actions of the demonstrations, so the return of the demonstrations themselves could be added as a baseline/topline.
- How much leverage does the high-level policy actually have over the actions? The flow policy is invertible, but conditioning on current and next state might, depending on the environment, provide all required information to compute an action already. Some analysis on this aspect would be appreciated.

**Limitations:**

The paper is not clear on the quality of demonstrations that are expected for the method to work effectively.

**Strengths And Weaknesses:**

The paper is very well-written and easy to follow. Clarity in the experimental section could be improved by stating what the abbreviations in ablation studies mean, and the ablations in the appendix are a bit hard to follow. Overall, the idea presented is well-motivated, novel and concerns a topic of high interest. The technique mainly builds upon PARROT and includes tricks for fine-tuning on task-specific demonstrations.

The experiments clearly show that the proposed method is effective in the settings considered; in fact, there is no setting in which few-shot generalization does not already work much better than all baselines considered. Overall, this suggests a potential limitation that doesn't seem to be acknowledged explicitly in the paper: how much does the method rely on precise task-specific demonstrations being available? What would happen if the environment changes slightly, e.g., the microwave in the kitchen environment would be moved 10cm to the left? I'd be happy to raise my score if this limitation is properly addressed, as I think it would help greatly in judging the method's applicability and impact.

---

> ### Author Response · Authors · 2022-08-02
> **Response (1/2)**
>
> Thanks for valuable feedback.
>
> ### Q1a: Clarity of abbreviations
>
> Thanks a lot for this suggestion. We’ll add:
>
> **Table B1:** Revised abbreviations for the ablation study of CEIP. We changed hyphen (“-”) to plus (“+”) for consistency with notations like “TS+TA”
>
> | Method |  Task-specific flow | Explicit prior | Push-forward |
> |---|---|---|---|
> | CEIP | | |
> | CEIP+EX | |&check; |
> | CEIP+EX+forward | |&check; | &check;
> | CEIP+TS | &check; | |
> | CEIP+TS+EX | &check; | &check; |
> | CEIP+TS+EX+forward | &check; | &check; | &check;
>
> **Table B2:** Revised abbreviations for the ablation study of PARROT. “2way” and “4way” only appear in fetchreach ablation. See Fig. 11 for more
>
> | Method | Use task-agnostic data | Use task-specific data | Explicit prior | Push-forward |
> |---|---|---|---|---|
> | PARROT+TA | &check; | | | |
> | PARROT+TS | | &check; | | |
> | PARROT+(TS+TA) | &check; | &check; | | |
> | PARROT+TA+EX | &check; | | &check; | |
> | PARROT+TS+EX | | &check; | &check; | |
> | PARROT+(TS+TA)+EX | &check; | &check; | &check; | |
> | PARROT+TA+EX+forward | &check; | | &check; | &check; |
> | PARROT+TS+EX+forward | | &check; | &check; | &check; |
> | PARROT+(TS+TA)+EX+forward | &check; | &check; | &check; | &check; |
> | PARROT+2way+TS | part of (see Fig. 11) | &check; | | |
> | PARROT+4way+TS | part of | &check; | | |
> | PARROT+2way | part of | | |
>
>
> ### Q1b: Clarity of ablation
>
> The ablation in the appendix is divided into two subsections, one for fetchreach and one for kitchen (ablation for office is in the main paper).
>
> Fetchreach ablation (Sec. D.1) studies five questions:
>
> **1. What’s the effect of each component (task-specific flow, explicit prior and push-forward) in CEIP for fetchreach?** The answer: “unnecessary for a simple environment,” as discussed in L740-744 and Fig. 9
>
> **2. Does the number of flows in CEIP affect results?** The answer: “more flows improve results,” as discussed in L745-747 and Fig. 10
>
> **3. What’s the effect of explicit prior and push-forward technique for PARROT?** The answer: “unnecessary for a simple environment,” as discussed in L753-755 and Fig. 11
>
> **4. Will hand-picked more relevant task-agnostic data help PARROT?** The answer: yes, as discussed in L755-756 and Fig. 11
>
> **5. What do the generated trajectories look like for each method, before and after RL training?** See Fig. 12 for the former and Fig. 13 for the latter. The trajectories show: CEIP works the best
>
> Kitchen ablation (Sec. D.2) studies four questions:
>
> **1. Is RL useful when the initial reward is already maximal?** The answer: yes in order to reduce episode length, as discussed in L771-773 and Fig. 14
>
> **2. What is the effect of each component in CEIP?** The answer: “TS and EX are both crucial”, as discussed in L775-778 and Fig. 15
>
> **3. What is the effect of each component in PARROT?** The answer: “explicit prior helps”, as discussed in L780-782 and Fig. 16
>
> **4. What is the coefficient for the task-specific flow?** The answer: “the mixture doesn’t degenerate to one flow”, as discussed in L783-793 and Fig. 17
>
> ### Q2: “method mainly builds upon PARROT and includes tricks for fine-tuning”
> We kindly disagree. While we use the PARROT backbone, there are two major differences between our method and PARROT. Both lead to significant improvements and new directions of research:
>
> **1. Normalizing flow mixture.** Our linear combination improves results and never appears in prior work. As shown in Fig. 5c and 16, compared with PARROT equipped with explicit prior, our mixture significantly improves the initial reward, thus CEIP converges faster. Prior works on normalizing flow mixtures use Gaussian mixture latent distribution [3], variational mixture [4], boosting [5], relaxation of invertibility [6] etc.; however, almost all prior works [4, 5, 6, 7, 8] only use simple datasets (e.g. MNIST / synthetic). [3] addresses point-cloud generation and reconstruction, but aims for stronger expressivity instead of distilling knowledge from different tasks. To our best knowledge, our CEIP flow mixture is the first mixture-of-normalizing-flow work that distills knowledge from multiple tasks in an RL setting, using a never-before-studied combination method.
>
> **2. The explicit prior.** Our explicit prior significantly improves behavior cloning (Tab. B5), CEIP (Fig. 15) and PARROT (Fig. 5c). Though explicit prior was studied recently (L348) in robotics, this component hasn’t found its way to RL literature (L346-348); we adopt it and propose a novel pushforward technique.
>
> We hence think CEIP goes beyond “PARROT + tricks for fine-tuning.”

---

> ### Author Response · Authors · 2022-08-02
> **Response (2/2)**
>
> ### Q3: How much does the method rely on precise task-specific demonstrations?
>
> Our experiments follow prior work (SKiLD [1] and FIST [2]), which don’t discuss precision of task-specific data. FIST has a very short discussion on noise in task-agnostic data and concludes with non-robustness to noise.
>
> To fill this gap, as suggested by the reviewer, we move the items in the office environment for CEIP+TS+EX and CEIP+TS+EX+forward. We choose the office environment instead of the suggested kitchen environment because the former can be changed with one line of code, while the latter would require coordinating the positions of multiple components in xml files (e.g., microwave hitbox, microwave walls, microwave door, microwave door handle and the corresponding goal).
>
> The original office environment uses a [-0.01, 0.01] uniformly random noise for the starting position of each dimension for each item in the environment. We increase this noise at test time (which the agent never sees in imitation learning) and show the result in Tab. B3. Albeit an improvement on FIST, CEIP is still not robust to imprecise demonstrations, which is a limitation we will add into the discussion.
>
>
> **Table B3:** Results of CEIP+TS+EX, CEIP+TS+EX+forward and FIST with random positioning of items
>
> Noise level | CEIP+TS+EX | CEIP+TS+EX+forward | FIST
> ---|---|---|---
> 0.01 (original) | 4.17 | 6.33 | 5.6
> 0.02 | 4.20 | 4.17 | 3.8
> 0.05 | 0.57 | 0.83 | 0.6
> 0.1 | 0.05 | 0.1 | 0.1
> 0.2 | 0.01 | 0.02 | 0
>
> ### Q4: How well will replaying existing demonstrations work?
>
> Below is the result for replaying the task-specific demonstration, averaged over 3 runs. We observe that the replay can’t solve the task, but works decently in some cases.
>
> **Table B4:** Results of replaying existing demonstrations (mean reward and std.dev); we list CEIP+TS+EX+forward (after RL) for convenience
>
> Environment | Replay | CEIP+TS+EX+forward
>  --- | --- | ---
> Kitchen-SKiLD-A | 1.0(+-0.82) | 4.0(+-0.00)
> Kitchen-SKiLD-B | 0.67(+-0.94) | 3.93(+-0.08)
> Kitchen-FIST-A | 2.33(+-0.47) | 3.95(+-0.05)
> Kitchen-FIST-B | 0.67(+-0.47) | 3.89(+-0.07)
> Kitchen-FIST-C | 2.33(+-0.94) | 3.92(+-0.06)
> Kitchen-FIST-D | 2.33(+-0.94) | 3.94(+-0.07)
> Office | 4.67(+-0.83) | 6.33(+-0.30)
>
> ### Q5: Is the current and next state sufficient for computing the optimal action?
>
> They aren’t. The result listed in Tab. B5 suggests that knowing the current and the next states improves policies. However, results are not as good as CEIP and exhibit a large variance.
>
> **Table B5:** Results of behavior cloning (mean reward and std.dev); we list CEIP+TS+EX+forward (after RL) for convenience
>
>  Environment | BC | BC+EX | BC+EX+forward | CEIP+TS+EX+forward
>  --- | --- | --- | --- | ---
>  Kitchen-SKiLD-A | 0.02(+-0.04) | 1.52(+-1.15) |  2.2(+-0.62) | 4.0(+-0.00)
>  Kitchen-SKiLD-B | 0.03(+-0.08) | 1.03(+-0.90) | 0.8(+-0.75) | 3.93(+-0.08)
>  Kitchen-FIST-A | 0.67(+-0.76) | 2.17(+-0.06) | 3.03(+-0.15) | 3.95(+-0.05)
>  Kitchen-FIST-B | 0.4(+-0.59) | 2.13(+-0.47) | 1.87(+-0.29) | 3.89(+-0.07)
>  Kitchen-FIST-C | 0.5(+-0.75) | 2.2(+-1.61) | 1.9(+-0.96) | 3.92(+-0.06)
>  Kitchen-FIST-D | 0.67(+-0.39) | 1.63 (+-1.42) | 2.17 (+-1.67) | 3.94(+-0.07)
>  Office | 0.62(+-0.59) | 0.53(+-0.42) | 1.83(+-0.49) | 6.33(+-0.30)
>
> **References**
>
> [1] K. Pertsch et al. Demonstration-guided reinforcement learning with learned skills. CoRL, 2021
>
> [2] K. Hakhamaneshi et al. Hierarchical few-shot imitation with skill transition models. ICLR, 2022
>
> [3] J. Postels et al. Go with the Flows: Mixtures of Normalizing Flows for Point Cloud Generation and Reconstruction. 3DV, 2021
>
> [4] G. Pires et al. Variational Mixture of Normalizing Flows. arXiv, 2020
>
> [5] R. Giaquinto et al. Gradient Boosted Normalizing Flows. NeurIPS, 2020
>
> [6] R. Cornish et al. Relaxing Bijectivity Constraints with Continuously Indexed Normalising Flows. ICML, 2020
>
> [7] L. Dinh et al. A RAD approach to deep mixture models. ICLR Workshop, 2019
>
> [8] P. Izmailov et al. Semi-Supervised Learning with Normalizing Flows. ICML, 2020

---

### Official Review · Reviewer_N6m9 · 2022-07-11

**Rating:** 5
**Confidence:** 4
**Soundness:** 2 fair
**Presentation:** 2 fair
**Contribution:** 2 fair

**Summary:**

The paper presents a technique for combining "explicit" and "implicit" priors for reinforcement learning with demonstrations. Implicit priors refer to priors that store the knowledge from demonstration in a neural network of some kind (such as a flow-based generative model), while explicit priors store demonstrations as a database that can be queried when learning a new task. The overall method has several steps:

1. Given a dataset of task-agnostic (TA) demonstrations, cluster them using k-means (where the last state of each trajectory is used as the clustering feature). If the given datasets are already divided into separate tasks, this step can be skipped.
2. Learn a flow-based prior for each of the cluster from the previous step, and learn a flow-based prior for task-specific trajectories. The number of task-specific trajectories is much less than the total number of trajectories. Often, there's only one task-specific trajectory (one-shot learning).
3. Learn weights to combine the numerous flow-based models into a single model. These weights are learned by minimizing a loss on  the task-specific dataset, and I have some questions about this step below.
4. Given a new task, train a policy \pi(z|s) that controls the prior model, which in turn controls the environment. This is similar to prior work (PARROT).
5. You are also allowed to access a dataset of expert trajectories on the current task. This dataset is accessed via a retrieval procedure wherein you search for a state that is most similar to the current state, and retrieve the next state. The next state is then passed as an additional input to the flow-based models, allowing the flow-based model to act as a sort of inverse dynamics model -- given state and next state, the flow based model predicts an action. This step is crucial for good performance from what I can tell.

The authors evaluate their method on simulated robot manipulation tasks.

Currently, my rating for the paper is somewhat low as I am not able to make sense of the results, and I am hoping the authors are able to answer the questions I asked here so that I can revise my rating accordingly.

Post-rebuttal: I have updated my rating to a borderline accept following the author response, as the authors have addressed a most of my concerns. I would encourage the authors to release their code if they can, as it would make the paper reproducible, and would spur future research in this area. One of my concern was that the results seemed almost too good to be true, and this concern can also be put to rest with a code release.

**Questions:**

- Authors use the terms “explicit priors” and “implicit priors” frequently, but these terms are never clearly defined (and they are not standard terms, as far as I know). From my reading, it seems like implicit priors refers to a method that trains a neural network using provided demonstrations, while explicit priors store all demonstrations in memory and can be later accessed using a nearest neighbor style technique. Can you clarify what exactly these two terms mean, and add these definitions towards the beginning of the paper?
- The authors train several different flows on the task-agnostic and task-specific datasets, and then combine these flows using some coefficients, and these coefficients are learned by minimizing the log-likelihood of the “combined flow” on task-specific trajectories. However, given that one of the flow models was trained on task-specific trajectories alone, this subsequent optimization might result in the coefficient for the task-specific flow model to receive most of the weight. Am I missing something here? Can authors share what coefficients were assigned to the task-specific flow model in their experiments?
- I understand the motivation behind the push-forward term in the retrieval objective function (Equation 6), but I did not completely follow the definition of the indicator function \delta. Can the authors add another equation that precisely defines when \delta = 1? Is it when s_next = s?
- In Figure 3, why does CEIP have a much higher reward right at the start of training (when steps = 0)? Why do other flow-based methods (such as PARROT) start with a much lower return? In particular, I would expect PARROT-TS to have a good initial performance, given that it’s trained on data for the task at hand (and task is relatively simple). Also, since CEIP trains on both TA and TS data, it would make sense to compare against a version of PARROT that uses both TA and TS.
- In Figure 4, the learning curve for CEIP for about half the tasks are essentially flat, with maximum performance achieved without any finetuning. Can the authors explain what is happening here? Is imitation learning alone sufficient for solving this task? The results seem very puzzling to me.


**Limitations:**

There is almost no discussion on limitations in the paper (the authors only briefly mention the computational cost of training many different flow models), which is disappointing. The lack of discussion around limitations, combined with results that seem almost too good to be true is a bit concerning.

**Strengths And Weaknesses:**

Strengths
- The paper tackles an important problem: how to best make use of task-agnostic and task-specific demonstrations to speed up reinforcement learning of new tasks.
- The paper obtains some impressive results on an interesting suite of tasks.
- The paper compares against a number of baselines, competing methods, and ablations.

Weaknesses
- The overall method is somewhat complex, with a number of moving pieces, and some of the design choices (such as the retrieval system) seem a bit ad-hoc.
- The experimental results are a bit puzzling, and I will add more details on this in the Questions section below.
- Most of the analysis / ablations are actually in the Appendix. Overall, the authors spend too much space explaining the method (which could have been shortened, since we don't need all the details on how flow-based models are trained, we can refer to prior work for that), and too little space analyzing the experimental results.

---

> ### Author Response · Authors · 2022-08-02
> **Response (1/2)**
>
> Thanks for valuable feedback.
>
> ### Q1a: Complexity of method and number of moving parts
>
> Our method is simpler to train and has no more moving parts than prior work that combines task-specific and task-agnostic data.
>
> E.g., SKiLD [1] trains 1) a VAE with LSTM; 2) two separate prior models that mimic the VAE sequence encoder, one for task-agnostic and one for task-specific data; 3) a binary classifier determining which prior to use; and 4) an RL agent with reward shaping whose coefficient requires to be learned. FIST [2] trains 1) an explicit prior (the retrieval system) using contrastive learning, 2) an LSTM-VAE similar to SKiLD; and 3) the implicit prior structure similar to SKiLD.
>
> In contrast, our method consists of 1) a non-parametric preprocessing (k-means); 2) training of simple single-layer flows (the mixture is also a single-layer flow); and 3) an RL agent independent of the flow architecture.
>
> ### Q1b: Ad-hoc design choices for retrieval
>
> We used a simple yet effective retrieval. We are excited to see that it yields good results. Surely, a more elaborate system can further improve results. We leave this to future work.
>
> ### Q2: Too much space for explaining methods and too little for analyzing results
>
> Thanks for the suggestion. We’ll defer details to the appendix and compress method explanation in the paper. We’ll use the space to move ablations from appendix to paper.
>
> ### Q3: Definition of “Implicit Priors” and “Explicit Priors”
>
> Both were explained early in the paper. We defined “implicit prior” in L24-25: “... distills the information within the demonstrations into an implicit prior by encoding available demonstrations into a deep net.” We defined “explicit prior” in L30-32: “enable the agent to maintain a database of demonstrations, which can be used to retrieve state-action sequences given an agent’s current state.” This is consistent with the reviewer’s understanding. We’ll clarify.
>
> ### Q4: Do results overly rely on task-specific flow? What coefficients were assigned to the task-specific flow?
>
> Results don't overly rely on the task-specific flow. We empirically studied this in Fig. 17 in the appendix. As stated in L784-786, if the result overly relies on the task-specific flow, the coefficient \mu_{n+1} for the task-specific flow would be 1 and \mu_i for all other flows would be 0. However, in Fig. 17, the coefficient for the task-specific flow (orange curve) is far from 1 (below 0.05), and the coefficient for a particular task-agnostic flow (blue curve) is far from 0 (generally above 0.02).
>
> Intuitively, over-reliance in our design (Fig. 6 bottom) is discouraged because of the softplus function and the positive offset applied on \mu. For over-reliance, all task-agnostic flows f_i should have a coefficient of \mu_i=0, which is unreachable due to the positive offset of \mu, and hard to approach due to the softplus.
>
> ### Q5: Definition of \delta=1
>
> \delta is the indicator function. For a particular s_next in a trajectory \tau in the task-specific data, \delta=1 if and only if there exists a state s’_next in \tau, such that s’_next satisfies the following two properties: 1) s’_next is no earlier than s_next in \tau; 2) s’_next has been retrieved once in the same RL episode. s’_next is “earlier” than s_next if it has a smaller index in a trajectory \tau, sorted in ascending order of execution time. This imposes a monotonicity on the retrieved s_next, i.e., a state is hard to be retrieved twice, and it is hard to first refer to later steps in a trajectory and then go back to earlier ones. We’ll clarify and add the following formula:
>
> $$\delta(s_{next})=\begin{cases}1 (\text{if }\exists \tau\in D_{n+1}, s’_{next}\in\tau, \text{s.t. } s_{next}\in\tau, s’_{next} \text{ is no earlier than } s_{next} \text{ and has been retrieved}) \\\\0 (\text{otherwise})\end{cases}$$
>
> where $\tau$ is a trajectory and $D_{n+1}$ is the task-specific data.
>
> ### Q6a: Higher reward at start in Fig. 3
>
> The higher initial reward of CEIP in Fig. 3 highlights the robustness of CEIP to starting point randomization in fetchreach. Fig. 8 shows 3 ways of starting point randomization. All results in the paper use the most challenging randomization illustrated in Fig. 8c. We find PARROT to work well with simpler randomization methods (Fig. 8a,b), but to struggle with the challenging randomization shown in Fig. 8c. In contrast, CEIP works well regardless of starting point randomization.

---

> ### Author Response · Authors · 2022-08-02
> **Response (2/2)**
>
> ### Q6b: Comparison to PARROT with task-specific and task-agnostic data
>
> Great suggestion. The result on fetchreach was listed in Fig. 11, which is no better than other variants of PARROT. For kitchen and office we conduct new experiments, and contrast this variant of PARROT (denoted as PARROT+(TS+TA)) to CEIP (numbers reproduced from Fig. 4 and 5). For both tables, we also list CEIP with task-specific flow, explicit prior and pushforward technique (CEIP+TS+EX+forward) for convenience.
>
> **Table A1:** Results of PARROT+(TS+TA) (reward) before RL training
>
> Env | PARROT+(TS+TA) |  CEIP+TS+EX+forward
> ---|---|---
> Kitchen-SKiLD-A | 1.29  | 4
> Kitchen-SKiLD-B | 0 | 3.32
> Kitchen-FIST-A | 0 | 3.24
> Kitchen-FIST-B | 0 | 3.75
> Kitchen-FIST-C | 0.5 | 3.85
> Kitchen-FIST-D | 1 | 3.9
> Office | 0 | 3.55
>
> **Table A2:** Results of PARROT+(TS+TA) (reward) after RL training
>
> Env | PARROT+(TS+TA) | CEIP+TS+EX+forward
> ---|---|---
> Kitchen-SKiLD-A | 1.56 | 4
> Kitchen-SKiLD-B | 1.07 | 3.93
> Kitchen-FIST-A | 1.77 | 3.95
> Kitchen-FIST-B | 0 | 3.89
> Kitchen-FIST-C | 0.93 | 3.92
> Kitchen-FIST-D | 2.55 | 3.94
> Office | 0 | 6.33
>
> Studying both tables, we find that PARROT+(TS+TA) does not work.
>
> ### Q7a: Why are curves in Fig. 4 flat?
>
> As the reviewer correctly stated, CEIP reaches the maximum reward before RL, and continues to solve all subtasks throughout it. This is because the flow-mixture with the help of an explicit prior and a task-specific flow can transform a policy close to a normal distribution to solve all subtasks. This is achieved as the flow learns to transform a normal distribution into an expert policy. For this, both the explicit prior conditioning on s_next and the task-specific single flow are important. The curve won’t be flat without either component.
>
> Also note, the agent improves during RL training despite the flat reward curve: as shown in Fig. 14, the average episode length decreases during RL training, increasing the discounted reward.
>
> ### Q7b: Is imitation learning sufficient for solving the task?
>
> No. Imitation learning alone, e.g., behavior cloning, is brittle as shown below:
>
> **Table A3:** Results of behavior cloning (mean reward and std. dev); we also list CEIP+TS+EX+forward (after RL) for convenience
>
> | Env | BC | BC+EX | BC+EX+forward | CEIP+TS+EX+forward |
> | --- | --- | --- | --- | --- |
> | Kitchen-SKiLD-A | 0.02(+-0.04) | 1.52(+-1.15) |  2.2(+-0.62) | 4.0(+-0.00) |
> | Kitchen-SKiLD-B | 0.03(+-0.08) | 1.03(+-0.90) | 0.8(+-0.75) | 3.93(+-0.08) |
> | Kitchen-FIST-A | 0.67(+-0.76) | 2.17(+-0.06) | 3.03(+-0.15) | 3.95(+-0.05) |
> | Kitchen-FIST-B | 0.4(+-0.59) | 2.13(+-0.47) | 1.87(+-0.29) | 3.89(+-0.07) |
> | Kitchen-FIST-C | 0.5(+-0.75) | 2.2(+-1.61) | 1.9(+-0.96) | 3.92(+-0.06) |
> | Kitchen-FIST-D | 0.67(+-0.39) | 1.63 (+-1.42) | 2.17 (+-1.67) | 3.94(+-0.07) |
> | Office | 0.62(+-0.59) | 0.53(+-0.42) | 1.83(+-0.49) | 6.33(+-0.30) |
>
> ### Q8: Discussing limitations
>
> We thank the reviewer for pointing out the need to better discuss limitations beyond mentioning computation in L375-376. We’ll add the following:
>
> **Reliance on optimality of expert demonstrations.** Similar to prior work like SKiLD and FIST, our method assumes availability of optimal state-action trajectories for the target task. Accuracy of those demonstrations impacts results. Future work should improve robustness and generality.
>
> **Balance between the degree of freedom and generalization in fitting the flow mixture.** Fig. 9 reveals that more degrees of freedom in the flow mixture improve results of CEIP. Our current design uses a linear combination which offers O(n) degrees of freedom (\mu and \lambda), where n is the number of flows. In contrast, too many degrees of freedom causes overfitting. It’s interesting future work to study this tradeoff.
>
> We invite the reviewer to discuss further limitations which we are happy to include.
>
> ### Q9: Are the results too good to be true?
>
> The results aren’t too good to be true.
>
> 1. Our improvements have been validated to be consistent across three benchmarks with systematic ablations.
>
> 2. The key of our strong results are due to our **combination** of **1-layer flows** with **explicit prior**, which are missing in the baselines. SKiLD and FIST have an LSTM-VAE architecture, which is too heavy with few task-specific trajectories compared to 1-layer flows; PARROT includes neither explicit prior nor flow combination.
>
> 3. We witness great performance improvement upon introducing our components into the baselines. PARROT (Fig. 5c,16) and behavior cloning (Tab. A3) with explicit prior work much better; compared to PARROT with explicit prior, CEIP converges faster with the combination of flows (in Fig. 5c and 16, PARROT+EX works well but converges slower). This again proves the effectiveness of our design.
>
> **References**
>
> [1] K. Pertsch et al. Demonstration-guided reinforcement learning with learned skills. CoRL, 2021
>
> [2] K. Hakhamaneshi et al. Hierarchical few-shot imitation with skill transition models. ICLR, 2022.

---

> ### Comment · Reviewer_N6m9 · 2022-08-04
> **Response to author response**
>
> Thank you for your response.
>
> Here are some further questions/suggestions/clarifications:
>
> 1. Do authors have any intuitions on why the weights assigned to task-specific flows are small? Given that the procedure for fitting the task-specific flow and the procedure for determining weights optimizes the same objective (negative log-likelihood on task-specific trajectories), I wonder why the weights assigned to task specific flows are this small.
> 2. The definition of the delta function is helpful. Please add it to the paper. Neurips allows you to update the pdf.
> 3. Thanks for the clarification regarding mean reward / episode length. I think mean (undiscounted) return is a better metric to plot on RL learning curves (instead of the mean reward that is currently plotted), and is more widely used when comparing RL algorithms (for example, see [1]). Can the authors update the relevant plots with mean return instead, so that we can better see the impact of RL?
> 4. When I said imitation learning alone, I was referring to CEIP before any RL, not vanilla behavior cloning.
>
> I will update my rating after the next response.
>
> [1] Haarnoja et al., Soft Actor-Critic: Algorithms and Applications, 2018.

---

> > ### Author Response · Authors · 2022-08-06
> > **Response on Further Questions**
> >
> > ### Q1: Why are the weights assigned to task-specific flows so small?
> >
> > Note, in the kitchen and office environment there are 24 task-agnostic flows and 1 task-specific flow. A uniform distribution assigns a weight of 1/25 = 0.04 to each flow. As we have mentioned in the last response, the architecture encourages the coefficients to be closer to a uniform distribution so as to avoid over-reliance on one flow.
> >
> > ### Q2: Adding the definition of delta function to the main paper
> >
> > Thanks a lot for the suggestion. We have modified the main paper to add this definition as suggested (marked blue in the updated pdf), as well as all suggestions mentioned in the original review. We are sticking to the 9-page limit for now but will move important experiments to the main paper for the camera-ready version upon acceptance where we have a 10-page limit.
> >
> > ### Q3:  Can the authors update the relevant plots with mean return instead, so that we can better see the impact of RL?
> >
> > In the original submission, we plotted every figure of the paper to show the undiscounted mean reward. The “discounted return” is only the reason why we observe a “shortened episode length”. In the original Fig. 14, we show the episode length, which is a surrogate for discounted return because undiscounted return, as shown in the paper, is already at its maximum and doesn’t reveal the improvements made by RL.
> >
> > ### Q4: Is CEIP before any RL sufficient for solving the tasks?
> >
> > Thanks a lot for the clarification. As suggested by Fig. 4 and 5, CEIP before any RL is sufficient for solving some of the kitchen environments (but not all), and is not sufficient for solving the office environment (where the maximum reward is 8). In the cases where CEIP before RL can solve the task, RL shortens the episode length and refines the policy.

---

> > > ### Comment · Reviewer_N6m9 · 2022-08-08
> > > **Thanks for the response; rating updated.**
> > >
> > > Thanks for the response. I have updated my rating accordingly.

---

### Author Response · Authors · 2022-08-06
**Revised Version of Our Main Paper and Appendix**

We thank all reviewers for their valuable and insightful comments. We have updated the pdf which integrates all advice and all new experiments conducted in response to all reviewers. We highlight all modified parts using blue font.

---

### Meta-Review · Area_Chair_8cb3 · 2022-08-24

**Recommendation:** Accept
**Confidence:** Less certain

**Metareview:**

All three reviewers have elected to accept the paper, with accept ratings of 5,6,7.

The reviews were thorough and demonstrated an understanding of the paper, and the authors have addressed many of the suggested edits. I like that the paper tackles the combination of parametric vs. non-parametric learning. One weakness of the paper, from a reproducibility POV (and also mentioned by the authors in limitations), is that there are a lot of moving pieces in the system (RL, non-parametric dataset lookup, one flow per task + 1 additional one for distilling them). It would seem quite annoying to implement correctly, if starting from scratch (but this is just an aesthetic feedback).

Despite the authors saying that the paper is "not too good to be true", I still find the stark contrast between baselines and the proposed method a bit hard to believe. I believe the code (if released) by the authors would reproduce the stated results in the paper, but what I am more skeptical of is that the baselines couldn't be tuned to perform much better. This is important for this specific paper, given the complexity of the method: a practitioner would want to know whether there is a simpler way to implement the improvements proposed here. For example, authors mention "The key of our strong results are due to our combination of 1-layer flows with explicit prior, which are missing in the baselines. SKiLD and FIST have an LSTM-VAE architecture, which is too heavy with few task-specific trajectories compared to 1-layer flows; PARROT includes neither explicit prior nor flow combination."

This which makes me wonder whether there isn't some simpler way to implement this, i.e. k-NN retrieval paired with contrastive embeddings + small networks for behavior cloning.

A minor nit: The explicit / implicit priors terminology was also confusing to me, as I typically think of this as "amortized inference + retrieval" or "parametric learning + non-parametric learning".

Recommendation: accept.



**Award:**

No

---

### Decision · Program_Chairs · 2022-09-14

Accept